

# Waste animal fat with hydrothermal liquefaction as a potential route to marine biofuels

Efraim Steinbruch[1,*], Siddaq Singh[1,*], Maya Mosseri[1],
Michael Epstein[1], Abraham Kribus[2], Michael Gozin[3,4,5], Dušan Drabik[6]
and Alexander Golberg[1]

[1] Department of Environmental Studies, Tel Aviv University, Tel Aviv, Israel
[2] School of Mechanical Engineering, Tel Aviv University, Tel Aviv, Israel
[3] School of Chemistry, Tel Aviv University, Tel Aviv, Israel
[4] Center for Nanoscience and Nanotechnology, Tel Aviv University, Tel Aviv, Israel
[5] Center for Advanced Combustion Science, Tel Aviv University, Tel Aviv, Israel
[6] Agricultural Economics and Rural Policy Group, Wageningen University and Research, Wageningen, Netherlands
* These authors contributed equally to this work.

## ABSTRACT

Unused animal waste rendered fat is a potential feedstock for marine biofuels. In this work, bio-oil was generated using hydrothermal liquefaction (HTL) of nitrogen-free and low sulfur rendered bovine fat. Maximum bio-oil yield of 28 ± 1.5% and high heating value of 38.5 ± 0.16 MJ·kg$^{-1}$ was obtained at 330 °C at 50% animal fat solid load and 20 min retention time. The nitrogen and sulfur content were negligible, making the produced bio-oil useful marine biofuel, taking into account current stringent regulations on $NO_x$ and $SO_x$ emissions. The economic analysis of the process, where part of the bovine fat waste is converted to the bio-oil and the semi-solid residues can be used to supply the heat demand of the HTL process and alternately generate electricity, showed that our process is likely to generate a positive profit margin on a large scale. We also showed the growing economic importance of electricity in the revenues as commercial production becomes more energy efficient.

## INTRODUCTION

The urgent need to replace 229 million metric tons of marine fuels derived from fossil sources annually is crucial for achieving society's decarbonization goals and ensuring long-term sustainability in the transport sector (*IMO, 2019*). Biofuels have emerged as one of the viable pathways to replace fossil fuels in the marine industry (*Hansson et al., 2019*; *Mukherjee, Bruijnincx & Junginger, 2020*; *Zhou et al., 2020*). However, the widespread adoption of biomass conversion technologies for energy production still faces challenges related to reactor efficiency, process costs, and the sustainability of biomass feedstocks (*Hansson et al., 2019*; *Mukherjee, Bruijnincx & Junginger, 2020*; *Zhou et al., 2020*).

Corresponding author
Alexander Golberg,
agolberg@gmail.com

Recent evidence suggests that hydrothermal liquefaction (HTL) utilizing sub-critical water at moderate temperatures (typically around 300 °C) and high pressure offers a promising solution for converting wet biomass into crude-like bio-oil. This technology shows potential for generating sustainable biofuels applicable to land transportation, aviation, and the marine sector (*Hsieh & Felby, 2017*). However, designing a scalable HTL process presents significant challenges, including selecting the appropriate feedstock and determining optimal process conditions, which can vary substantially depending on the specific feedstock (*Gollakota, Kishore & Gu, 2018*).

One underexplored source for HTL-derived bio-oils is waste animal residues, which currently represent an unavoidable component of the food supply chain. By leveraging these waste materials, the HTL process has the potential to transform them into valuable biofuels, thereby addressing both environmental and waste management concerns.

Further research and development efforts are needed to optimize the HTL process using waste animal residues as feedstock. This includes exploring the characteristics and properties of bio-oils derived from different animal residues, evaluating the efficiency and cost-effectiveness of the HTL process, and assessing the environmental sustainability of utilizing these biofuels in the marine sector.

By tapping into this unexplored potential, the utilization of waste animal residues in HTL-derived biofuels could contribute significantly to the decarbonization of the marine industry while simultaneously addressing waste management challenges in the food supply chain.

Global meat production reached 315 million tons in 2013, with beef and buffalo accounting for approximately one-fifth of this total (*Bruinsma, 2017*; *Ritchie & Roser, 2019*). According to the Food and Agriculture Organization of the United Nations (FAO), meat production is projected to grow at a rate of 1.5% *per annum* between 2015 and 2030 (*Bruinsma, 2017*). However, the production of animal meat is accompanied by significant waste generation, with cattle, sheep, and lambs resulting in 47% waste, pigs contributing 44% waste, and poultry generating 37% waste (*Adhikari, Chae & Bressler, 2018*). This waste includes various animal by-products (ABP) such as fat trim, meat, viscera, bone, blood, and feathers (*Hamilton, 2004*), the proportions of which vary depending on the efficiency of the processing plant. Finding innovative solutions to utilize this waste is crucial for diverting it from landfills and reducing greenhouse gas emissions (*Bitnere & Searle, 2017*).

Due to the environmental concerns associated with animal carcasses and other by-products from the meat slaughter industry, many countries in the European Union (EU) are supporting the production of biofuels from animal fat (*Emissieautoriteit, 2018*). Furthermore, the utilization of waste animal fat can address feedstock cost challenges in biofuel production, making it an environmentally and economically promising source of energy.

Currently, ABPs are typically processed through rendering, where they are collected, ground, and heated at temperatures ranging from 115 °C to 145 °C until the moisture content is sufficiently reduced, and fat and proteins are separated, resulting in products such as meat and bone meal, animal fat, and effluent water (*Hamilton, 2004*; *Walsh, 2014*).

Cattle fat obtained from the rendering process, commonly known as tallow, accounts for approximately 10–15% of ABP. As the quantity of animal fat increases with meat production, converting this feedstock into biofuels presents an opportunity for sustainable utilization.

In the EU, rendered fat in 2011 was utilized for energy purposes (33%), animal feed (28%), and the production of soap and oleochemicals (22%). Biodiesel, mainly derived from tallow, accounted for two-thirds of the fat used for energy purposes. Transesterification, an industrial process, is commonly employed to convert oils and fats into biodiesel, generating fatty acid methyl esters (FAME) with similar ignition properties to conventional diesel oil. However, transesterification poses challenges such as the presence of free fatty acids and water content, which can affect catalyst efficiency and promote soap formation, hindering phase separation of esters and glycerol (*Feddern et al., 2011*). Moreover, the corrosiveness of the bases and acids used in transesterification can cause environmental issues (*Karmee, 2016*). Alternatively, hydrotreatment is another technique used to convert vegetable oils and animal fats into green paraffinic fuels. However, hydrotreatment requires a continuous supply of hydrogen, typically produced from fossil fuels, and expensive catalysts that can be sensitive to impurities in the feedstock (*Chiaramonti et al., 2016*; *Hájek et al., 2021*; *Sani et al., 2017*). These factors limit the resource efficiency and scalability of the hydrotreatment process. Solvent extraction has also been considered for valorizing waste fats, but many organic solvents used in this process pose health and environmental risks (*Li, Sakuragi & Makino, 2019*).

Pyrolysis is considered an appealing method for converting fat waste into bio-oil. However, the resulting oils often have a higher acidic value compared to the standards for fuel (*Ben Hassen-Trabelsi et al., 2014*; *Wiggers et al., 2009*). Additional studies mentioned in the review by *Su et al. (2022)* also highlight certain limitations of bio-oil for direct use as fuel. Although it has been proposed to use catalysts to reduce the acidic value (*Su et al., 2022*), this approach can increase the overall process cost, depending on the catalyst employed.

In a recent study comparing the resources and cost consumption between hydrothermal liquefaction (HTL) and pyrolysis for processing animal by-products, *Marcilla et al. (2019)* found that HTL, when combined with recycled aqueous phase, emerged as the most favorable process. Another study by *Chan et al. (2016)* revealed that fast pyrolysis of Malaysian oil palm empty fruit bunch had double the global warming potential (GWP) impact compared to HTL. Unlike the aforementioned methods, HTL is not affected by moisture content since water serves as the reaction medium. This makes HTL a suitable method for converting animal fat into liquid fuel.

HTL offers multiple benefits as it can handle feedstocks with varying moisture content. Since water is the reaction medium, a wide range of feedstocks can undergo HTL without the need for drying. Moreover, HTL is an environmentally friendly process that avoids the use of harsh chemicals. Additionally, catalysts are generally not required in HTL of biomass with a high content of lipids and proteins, further enhancing the promise of HTL for processing animal fats (*Lachos-Perez et al., 2022*).

The hydrolysis of animal fat has been known for two centuries, and it has been an industrial process since the late 1940s when Colgate-Emery patented the splitting of fat into fatty acids and glycerine. This fat-splitting process occurs at 250 °C and 50 bar (*Barnebey & Brown, 1948*). The hydrolysis mechanism of animal fat, or fat splitting in water, has been extensively studied (*Ackelsberg, 1958*; *Barnebey & Brown, 1948*; *Mills & McClain, 1949*; *Lascaray, 1949*; *Sturzenegger & Sturm, 1951*). However, these studies focused on achieving complete conversion to fatty acids and glycerine. One of their major conclusions was that hydrolysis of triglycerides and esterification are reversible reactions in equilibrium (*Ackelsberg, 1958*; *Sturzenegger & Sturm, 1951*).

In a recent study by *León, Marcilla & García (2019)*, ABP (meat waste before the rendering process) from bovine and porcine sources was subjected to hydrothermal liquefaction (HTL) to produce liquid fuel. The study reported a favorable bio-crude yield of 61% at a process temperature of 225 °C. The resulting product predominantly consisted of triglycerides and fatty acids. However, when higher temperatures were used, the bio-crude obtained contained amides and heterocyclic compounds, which could have adverse environmental effects (*León, Marcilla & García, 2019*).

Another recent study by *Yang et al. (2019)* investigated the effects of HTL on pork meat. The researchers achieved a maximum bio-crude yield of 55.6% at a higher temperature of 320 °C, using a 10% solid load (SL) and a 60-min retention time (RT). The bio-oil produced in this study contained fatty acids, hydrocarbons, amides, esters, and N-heterocyclic compounds, similar to the previous study but at elevated temperatures (*Yang et al., 2019*).

It is important to note that in both studies, the obtained bio-oil had a relatively high nitrogen content ranging from 2% to 5%. This nitrogen content can be attributed to the nitrogen (N) present in the raw material used. Biofuels with high nitrogen content require additional processing, such as denitrification, to meet emission standards and make the fuel usable while complying with regulations.

The present study aims to advance the waste hydrothermal liquefaction (HTL) process by utilizing feedstocks with low or no nitrogen content, such as rendered fat, instead of raw animal by-products (ABP). Co-products obtained from rendering, such as meat and bone meal (MBM) and processed animal protein (PAP), have higher economic value as animal feed rather than being converted into fuel.

To our knowledge, no prior study has described the subcritical water HTL process for converting rendered, nitrogen-free fat of bovine origin into liquid fuel without the need for any catalyst. Therefore, this study seeks to address this research gap by investigating the HTL of rendered bovine fat. Specifically, the study explores the influence of residence time and solid load on the yield of the resulting bio-oil.

The primary objective of this research is to generate a liquid fuel through HTL using rendered animal fat, which can be directly blended with marine fuels or transformed into valuable chemicals. The study also evaluates the impact of residence time and solid load on bio-oil yield. Additionally, fuel properties such as higher heating value (HHV), density, viscosity, total acid number, and elemental composition of the produced bio-oil are analyzed and compared with those of marine fuel.

This study surpasses laboratory-scale experiments and incorporates an initial economic analysis for a plant-scale system. The analysis integrates experimental data obtained from the new feedstock to simulate a large-scale system. Such an approach is novel and essential for assessing the feasibility of implementing the HTL process using rendered fat in real-world applications.

The experimental data and findings from this study have significant implications for the design and establishment of meat waste bio-refineries dedicated to marine fuel production. The information collected serves as a foundation for optimizing and scaling up the HTL process, leading to enhanced efficiency and economic viability of biofuel production from rendered animal fat.

## MATERIALS AND METHODS

### Animal fat waste

Celitron Ltd. (Vác, Hungary) provided the rendered bovine waste fat produced from animal by-products of the meat industry, using their version of the rendering process by Integrated Sterilizer and Shredder (ISS). In the ISS, the meat waste, which includes waste fat, skin, tendons, and bones, is subjected to mild temperatures in the range of 121–134 °C, for 34 min (*León, Marcilla & García, 2019*), followed by a decanter that separates the particulate matter (bone and protein fraction) from the liquid fraction. Finally, the fat was separated from the liquid phase. The animal fat (AF) was stored under refrigerated conditions (4 °C) between experiments. The AF was mixed with deionized water (DW) at different proportions, as the feedstock for the HTL process.

### Hydrothermal liquefaction

Hydrothermal treatment of the AF was conducted in a batch reactor. The experimental system (Fig. S1) consists of a 250 mL batch reactor heated by an electric heater (model CJF-0.25; Keda Machinery, Foshan City, China). The temperature was controlled and measured with an MRC TM-5005 digital temperature gauge, using a "1/16" thermocouple type K (Watlow, St. Louis, MO, USA). The heating rate was about 5 °C-min$^{-1}$. The pressure inside the reactor was constantly measured by a pressure gauge (model PS-9302; MRC, Beverly Hills, CA, USA) with a 50-bar sensor (model PS100-50BAR; MRC, Beverly Hills, CA, USA). A mixer with a magnetically coupled drive was set at 70 RPM to mix the slurry inside the reactor. The magnetic coupling was water-cooled by a chiller (Cw-5200ai; Guangzhou Teyu Electromechanical Co., Ltd, Guangzhou, China). The reactor has two gas sampling ports and one liquid sampling line. Before entering the sampling tube the line includes a condenser and a cold trap. The gas samples were collected with a one-liter sampling bag. Before reaching the gas sampling bag, the gas sampling passed through a condenser and a cold trap.

For each experiment, the reactor was filled with 100 grams mixture of feedstock and distilled water at different proportions. The experiments were conducted under the following process conditions. Three variations of solid load (SL) of 25%, 50% and 75% of AF in the reaction mixtures were tested at a temperature of 330 °C, with the residence time (RT) of 60 min. Multiple residence times (RT) of 20, 40, 60, 80 and 120 min were also

applied to the reaction mixtures at the temperature of 330 °C and AF's solid load of 50%. The miscibility of fatty acids derived from tallow in water occurs at 321 °C (*Mills & McClain, 1949*). A vacuum pump (operated at 0.13 mbar, ST-85; MRC, Beverly Hills, CA, USA) was used to remove air from the system before starting each experiment. The reactor was then heated to the selected temperature, and after a specific desired residence time, the reactor heater was switched off. The samples were collected after the reactor was cooled down to room temperature. No solvent was used to collect the oil from the reactor. All samples collected from the HTL reactor were subjected to centrifugation for phase separation. These samples were subjected to two rounds of centrifugation (30 min each) at 4,000 rpm in a swing-bucket rotor. The phase separation was favored by a swing rotor in comparison to a fixed angle rotor. Three phases were distinctly observed: bio-oil, solid-phase, and aqueous phase. The oil was collected from the top layer, after each round.

The bio-oil yield was calculated using Eq. (1).

$$[\text{Bio−oil Yield}](\%) = (W_{\text{bio−oil}})/(W_{\text{AF}}) \times 100\% \tag{1}$$

where $W_{\text{bio-oil}}$ is the weight of fat bio-oil collected after centrifugation and $W_{\text{AF}}$ is the weight of animal fat feedstock used.

## Ash and moisture analysis

For ash and moisture analysis, untreated feedstock samples and biochar were dried at 40 °C to constant weight and analyzed according to the D5142 method (ASTM D5142-90 (1998), DOI: 10.1520/D5142-90R98).

## Elementary analysis

Elemental (CHNS) analyses were performed at the Chemical and Surface Analysis Laboratory (Technion—Israel Institute of Technology, Haifa, Israel) using Flash2000 (Thermo Fisher Scientific CHNS Analyzer; Thermo Fisher Scientific, Waltham, MA, USA). The oxygen atom content was determined based on ASTM E870-82 by the following equation:

$$[\text{O}]\% = 100\% − ([\text{C}]\% + [\text{H}]\% + [\text{N}]\% + [\text{S}]\% + [\text{Ash}]\%) \tag{2}$$

## Density of bio-oil

The density of bio-oil samples was measured by using the DMA 35 Portable density meter at room temperature. Each sample was measured twice, and the mean value was determined. The value reported here is the mean of the duplicates.

## Higher heating values

Higher heating values (HVVs) of the bio-oil samples were obtained by using a bomb calorimeter (Parr 6200; Parr, Cleveland, OH, USA) equipped with a 1104 oxygen bomb (Parr, Cleveland, OH, USA) under an oxygen (30 atm) atmosphere. The bomb calorimeter was calibrated based on 1.0-g pellet of benzoic acid. For obtaining the caloric value of AF samples, 1.0 g of the untreated AF was analyzed according to ASTM D5865-13 (Standard Test Method for Gross Calorific Value of Coal and Coke).

## Fuel properties

Fuel properties were measured by Intertek AG (Switzerland). Density at 50 °C (EN ISO 12185), total acid number (ASTM D664), viscosity at 50 °C (ASTM D7042), viscosity at 80 °C (ASTM D7042), caloric value (ASTM D4809), water content (DIN 51777/1), elemental analysis for carbon (ASTM D5291), hydrogen (ASTM D5291), nitrogen (ASTM D5291). Trace elements were analyzed by using ICP-OES.

## Thermogravimetric analysis

Thermogravimetric (TG) analyses of AF samples were carried out on STA 449 F5 apparatus (Netzsch, Selb, Germany). Dry powder (2 mg) samples were subjected to analyses at a temperature range of 30–600 °C, with temperature ramping at a rate of $10\,°C\cdot min^{-1}$, under a nitrogen ($N_2$) atmosphere. The crucibles with the samples were sealed with a lid having a hole in the center.

## Infrared spectroscopy

Infrared spectroscopy analyses were performed using an FT-IR spectrophotometer (Tensor 27; Bruker, Billerica, MA, USA), equipped with a standard attenuated total reflectance (ATR) attachment (Pike). The samples were measured in the spectral range of $4,000–400\ cm^{-1}$ (at $4\ cm^{-1}$ resolution).

## Gas chromatography-mass spectrometry (GC-MS)

Gas Chromatography-Mass Spectrometry (GC-MS) analyses were performed with Agilent 5975 (Agilent Technologies, Santa Clara, CA, USA), using 122-5532UI: 1DB-5ms Ultra Inert capillary column (30 m length, 250 µm inner diameter, and 0.25 µm film thickness). The flow rate of the helium carrier gas was $1.2\ ml\cdot min^{-1}$. Samples of 1 µL were injected into an injector heated to 220 °C, with a split ratio of 9:1. The temperature program of GC oven included the initial step of 1.0 min at 80 °C, then ramping up to 310 °C, at the rate of $10\,°C\cdot min^{-1}$, and holding at the final temperature for 3 min. The mass spectral range in the GC-MS analyses was in the range of 50–500 amu. The compounds found were identified based on a comparison with the National Institute of Standards and Technology (NIST) mass spectra library (*NIST, 2021*). The analysis was performed by Intertek (Schweiz, Bern, Switzerland) AG, Switzerland.

## Statistical analysis

Statistical analysis was performed with Excel (ver. 13; Microsoft, Redmond, WA, USA) and data analysis package and RStudio (*RStudio Team, 2017*).

# RESULTS AND DISCUSSION

## Products of hydrothermal liquefaction

Figure 1A illustrates the HTL product separation after centrifugation. Three phases were distinctly observed in Figs. 1B–1D: bio-oil, aqueous phase and semi-solid phase, respectively. The bio-oil is a viscous light-to-dark brown oil. The solid phase consistency was similar to the untreated feedstock. The aqueous phase contained water-soluble

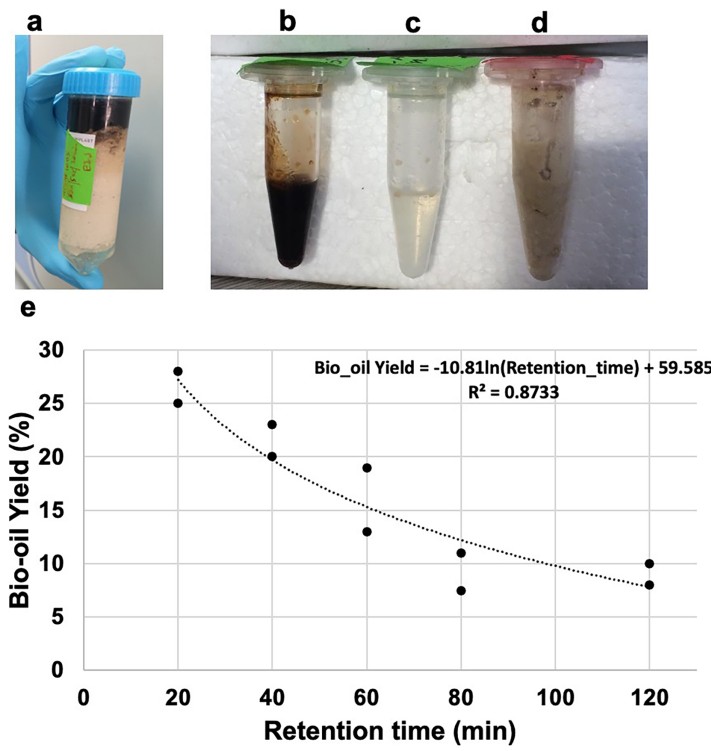

**Figure 1 Products from HTL of animal fat.** (A) Tube of HTL AF after centrifugation with three layers; (B) bio-oil; (C) aqueous phase; (D) semi-solid phase; (E) results obtained for bio-oil yield *vs.* retention time for experiments conducted at 330 °C and AF solid load of 50%. The model was fit the Excel (Microsoft, SE, ver 16.66.1 Solver).

organics and appeared as a transparent liquid. Although expected in the used temperature, we did not observe significant gas formation in the HTL process at this scale and with the used materials. The residual pressure in the reactor, after its cooling to room temperature, was <0.5 bar. This caused a meager inaccuracy in the mass balance calculation.

The increase in retention time did not have a significant effect on the density and the HHV of the bio-oil (Table 1): the densities were in the range of 0.896–0.899 g·cm$^{-3}$, and the high heating values were in the range of 37.35–38.84 MJ·kg$^{-1}$.

To the best of our knowledge, this is a first study on nitrogen-free rendered animal fat liquefaction. Previous works in the field of animal waste products were mostly focused on the whole animal tissues (Table 2). In most of the published articles on HTL (with more than 100 different tested feedstocks (*Gollakota, Kishore & Gu, 2018*)), the separation of the bio-crude is done using liquid solids extraction with solvents like dichloromethane or acetone (*Yang et al., 2019*). Such an approach, although potentially leads to higher bio-oil extracted yields (Table 2), requires significantly higher capital and operational investments in the plant infrastructure, as the solvents need recycling. The cost to benefit analysis comparison of mechanical *vs* chemical separation of bio-oils following HTL process is still needed before large scale implementation. However, using solvents for removing the residue from mechanical separation of moisture (6% in this study) could be a viable strategy to further improve the quality of the HTL products.

**Table 1 Retention time impact on the bio-oil yield.**

| Exp. # | Retention time (RT) (min.) | HHV (MJ·kg$^{-1}$) | Yield (wt %) | Density (g·cm$^{-3}$) |
|---|---|---|---|---|
| 1 | 20 | 38.67 | 28 | 0.898 |
| 2 | 20 | 38.44 | 25 | 0.898 |
| 3 | 40 | 38.91 | 23 | 0.988 |
| 4 | 40 | 38.77 | 20 | 0.898 |
| 5 | 60 | 38.70 | 14 | 0.899 |
| 6 | 60 | 37.90 | 19 | 0.900 |
| 7 | 80 | 38.25 | 8 | 0.898 |
| 8 | 80 | 38.35 | 11 | 0.898 |
| 9 | 120 | 36.45 | 8 | 0.897 |
| 10 | 120 | 38.24 | 10 | 0.896 |

Note:
HHV, yield, and density of bio-oils generated by HTL, as a function of the retention time (RT) at the temperature of 330 °C and AF's solid load of 50%.

Furthermore, it is imperative to highlight that our work has not focused on optimizing the mechanical separation process for bio-oils from water and residual fat. This optimization encompasses not only the reduction of energy and time inputs but also the maximization of separation efficiency, resulting in products with minimal moisture content and increased yield. A crucial consideration is that the hydrothermal liquefaction (HTL) of the initial fat could yield miscible products, which can still serve as viable biofuels. However, separating such miscible products solely through centrifugation may pose challenges, necessitating additional solvent extraction. This undertaking would entail conducting measurements to assess the miscibility of HTL products under various process conditions, quantifying the yield of both miscible and non-miscible bio-oil products, determining suitable supplementary separation methods, and conducting an economic evaluation to ascertain the feasibility of this approach.

## Retention time impact on the bio-oil yield

Experiments were conducted at temperature of 330 °C and pressure of 130 bars, with AF's solid load of 50%, applying multiple retention times of 20, 40, 60, 80 and 120 min, as this temperature was found to be optimal for waste animal products (*Yang et al., 2019*) and organic waste (*Saengsuriwong et al., 2021*). The effect of changing the retention time on bio-oil yield, while keeping the temperature and solid load constant, is illustrated in Fig. 1E and Table 1. The highest bio-oil yield of 28% was obtained at the shortest retention time of 20 min, while the lowest yield of 8% was observed at the longest retention time of 80–120 min. A decrease in bio-oil yield was observed with increasing residence time probably due to increased gasification or production of smaller molecules which were transferred to the aqueous phase (*Madsen & Glasius, 2019*). A similar trend was observed by *León, Marcilla & García (2019)*, who studied the HTL process using animal by-products. This trend was also observed by HTL processing of other biomasses, including animal carcass (*Yang et al., 2019*), cattle manure (*Yin et al., 2010*), macroalgae *Ulva prolifera* (*Yan et al., 2019*), *Cyanophyta* (*Guo et al., 2015*), and *Cunning lanceolata* (*Qu, Wei & Zhong, 2003*).

**Table 2 Bio-oil production through hydrothermal liquefaction of animal products. No catalyst, neutral pH.**

| Feedstock | Experimental parameters | Bio-oil | Reference |
|---|---|---|---|
| Animal by-products | Solid Load (%): 50%<br>Temperature (°C): 290<br>Residence time (min): 15<br>Bio-oil separation method: mechanical | Yield: 56.1%<br>HHV: 30.4 MJ kg$^{-1}$ | León, Marcilla & García (2019) |
| Streaky pork | Solid load (%): 10<br>Temperature (°C): 320<br>Residence time (min): 60<br>Bio-oil separation method: chemical (dichloromethane) | Yield: 55.6%<br>HHV: 42.2 MJ kg$^{-1}$ | Yang et al. (2019) |
| Streaky pork | Solid load (%): 20<br>Temperature (°C): 320<br>Residence time (min): 60<br>Bio-oil separation method: chemical (dichloromethane) | Yield: 58.6%<br>HHV: 39.7 MJ kg$^{-1}$ | Yang et al. (2022) |
| Food waste | Solid load (%): 12.5%<br>Temperature (°C): 340<br>Residence time (min): 30<br>Bio-oil separation method: chemical (acetone) | Yield: 56%<br>HHV: 37.33 MJ kg$^{-1}$ | Saengsuriwong et al. (2021) |
| Raw waste from porcine and bovine origin | Solid load (%): 50<br>Temperature (°C): 290<br>Residence time (min): 5<br>Bio-oil separation method: mechanical | Yield: 36.3<br>LHV: 29.1 MJ kg$^{-1}$ | Leon et al. (2018) |
| Processed animal protein (PAP) and fat (7:1) | Solid load (%): 50<br>Temperature (°C): 290<br>Residence time (min): 5<br>Bio-oil separation method: mechanical | Yield: 24.6<br>LHV: 31.6 MJ kg$^{-1}$ | Leon et al. (2018) |
| Rendered bovine waste fat | Solid load (%): 50<br>Temperature (°C): 330<br>Residence time (min): 20<br>Bio-oil separation method: mechanical | Yield: 28%<br>HHV: 38.67 MJ kg$^{-1}$ | This study |

However, the residence time at which the bio-yield started to decrease varied with the biomass origin, temperature, and other processing conditions. As there are chances of secondary and even tertiary reactions during the HTL, higher molecular weight intermediates can convert to lower molecular weight compounds, some of which could be liquids (Akhtar, Aishah & Amin, 2011) or gases (Madsen & Glasius, 2019). The decrease in bio-oil yield was also attributed to the degradation of bio-oil and re-polymerization at higher residence time (Akhtar, Aishah & Amin, 2011; Xue et al., 2016; Yang et al., 2019). Additional studies on the impact of lower temperatures on the HTL products and resulting economics will be beneficial.

The energy density (measured and calculated) of the untreated AF and the bio-oil, obtained at the maximum yield, their elemental analyses (CHNSO) and HHVs are shown in Table S1. The trace elements analysis of the bio-oil is shown in Table S2.

### Impact of solid load on bio-oil yield

Table 3 shows the bio-oil yields, HHVs, and densities, for different solid loads of feedstock. The bio-oil yields increased from 17% to 27%, while the solid loads increased from 50% to 75%. The bio-oil from the 75% solid load experiment also showed a slightly higher density of $0.91 \pm 0.02$ g·cm$^{-3}$ than all other bio-oil samples. It should be noted that the solid loads used in this study are much higher than the solid loads usually reported in the relevant literature (usually 5–25 w/v%, see in *Gollakota, Kishore & Gu (2018)*, *Madsen & Glasius (2019)*), potentially affecting the lower conversion rates that are reported in the literature about animal by-products. The highest solid load we found relevant for this study, was reported on animal by products and was 50% (*Leon et al., 2018*). Further possible increase of the solid load to 75% (Table 3) as reported here is of a significant practical importance, as it potentially could lead to the decrease of the HTL reactor size and thus costs and complexity of installation and maintenance.

*Xue et al. (2016)* showed that the response of bio-oil yield to the solid load could vary depending on the biomass. Other researchers reported that in the HTL study of hyacinth (*Singh et al., 2015*), bio-oil yield increased with increasing the biomass percentage, while in the HTL processing of cornstalk (*Liu, Li & Sun, 2013*) the yield decreased after reaching a maximum level, when the proportion of biomass was increased in the mixture. *Xue et al. (2016)* attributed the decrease in bio-oil yield to the competition between hydrolysis and re-polymerization reactions after the water/solids ratio crosses a specific limit. This observation may explain the trend of decrease in bio-oil yields, as the solid load was reduced from 75% to 25%.

One important observations in this study was that no bio-oil could be extracted from 25% solid hydrolysate of rendered fat when using our separation protocol. These results were in contrast to the findings reported by *Biller & Ross (2011)* and *Teri, Luo & Savage (2014)*, who observed a high yield bio-oil from HTL processing of sunflower oil at 10% and 15% of solid load, with a high reaction temperature of 350 °C.

The HHV also showed a slight increase from $38.3 \pm 0.4$ MJ·kg$^{-1}$, for 50% of the solid load, to $38.7 \pm 0.1$ MJ·kg$^{-1}$, for 75% of the solid load which is similar to HHV from other animal residues (Table 2). The processing of 75% of the solid load had the highest yield of 27% of bio-oil, which was similar to the yield from 50% of the solid load. In both processes, retention times were 60 min.

### Animal fat-derived bio-oil composition

The feedstock and bio-oils from various experiments were analyzed using DTG, FT-IR, and GC-MS to measure the composition of the bio-oil and the changes that occurred due to the HTL process.

**Table 3 HHV, yield and density of bio-oils generated at varying solid loads at the temperature of 330 °C and 60 min. RT.**

| Exp. # | Initial fat solid load (SL) (%) | HHV (MJ·kg$^{-1}$) | Bio-oil yield (%) | Density (g·cm$^{-3}$) |
|---|---|---|---|---|
| 11, 12 | 25 | _[1] | _[1] | _[1] |
| 5 | 50 | 38.70 | 14 | 0.899 |
| 6 | 50 | 37.90 | 19 | 0.900 |
| 13 | 75 | 38.77 | 27 | 0.905 |
| 14 | 75 | 38.60 | 25 | 0.904 |

**Note:**
[1] There was no extractable bio-oil in the hydrolysate from 25% SL.

## Thermogravimetric analysis

DTG and TGA thermograms of animal fat feedstock and bio-oils at various conditions are shown in Fig. S2. The thermal decomposition of the feedstock seems to start around 150 °C and gradually increases until 300 °C. This behavior was followed by a steeper curve as the temperature approached the 400 °C region, at which the main decomposition took place. A presence of small shoulder of degradation above 500 °C was also observed. The DTG thermogram of the AF feedstock followed a similar curve to a rendered fat, as reported by other authors (*León, Marcilla & García, 2019*; *Melzer et al., 2013*). Although the small shoulder was absent in the DTG thermogram reported by these authors, the main decomposition was observed around 400 °C. Leon described a big peak near 400 °C as decomposition of triglycerides, and an additional peak around 300 °C, corresponding to a lighter fraction, composed of free fatty acids (*León, Marcilla & García, 2019*). A similar conclusion was reached by *Melzer et al. (2013)*, as they conducted TG analyses on shea butter and jatropha oil, which also mainly contains triglycerides.

DTG thermograms of our bio-oils showed a shift in their peaks to a lower temperature, at around 300 °C, in comparison to the thermograms of the AF feedstock. As mentioned above, these lower temperature peaks represent lighter fractions. This observation indicates the transformation of triglycerides to lighter compounds in the HTL process. *León, Marcilla & García (2019)* also reported similar results, as their biocrude DTG thermogram at 290 °C closely resembles our thermogram of 50% of the solid load at 330 °C.

This observation was even clearer when we analyzed the thermogram for 75% of the solid load at the retention time of 60 min. In this case, we observed that the percentage of conversion from triglycerides to free fatty acids was lower, as the two major peaks were found to be in approximately 60:40 ratio, between near 300 °C and 400 °C. If we compare these results to the lower feedstock load of 50% with the same retention time of 60 min, the peak around 400 °C decreased, while a significantly larger peak could be observed around 300 °C, thereby showing a better conversion to lighter fraction for a lower percentage of the solid load. Similarly, *Yin et al. (2010)* observed that a higher mass ratio of cattle manure-to-water resulted in lower bio-oil yields. They asserted that this decrease in bio-oil yield could be due to self-condensation reactions resulting from the formation of solid residue.

Focusing on the shoulders of bio-oil thermograms for 50% of the solid load, between 350–450 °C, at the highest retention time of 120 min, we found the smallest area of peaks in comparison to the processes conducted with retention times of 20 and 60 min. This observation indicates a formation of a higher proportion of triglycerides, due to the increase in the retention time of the sample. However, the bio-oil yields were found to be lower in experiments with higher retention times, which indicates that the conversion products are diverted to the solid or to aqueous phases. Yang et al. (2019) reported similar observations while conducting HTL processes using the animal carcass as a feedstock. The TGA thermograms also highlighted the differences in the decomposition among AF's feedstock and different bio-oils.

## Infrared spectroscopy analysis

Figure S3 shows the FT-IR spectra of the animal fat feedstock compared to bio-oils obtained from HTL processes conducted under different parameter conditions. The graph shows that all the bio-oils follow the same trend, and their spectra overlap. However, in comparison to the feedstock, significant changes were observed between the feedstock and the bio-oils spectra, confirming a chemical transformation of animal fat occurred in the HTL process. The bio-oil's peaks have increased intensity as compared to the feedstock at frequencies of 3,007, 2,914, 2,851, and 719 $cm^{-1}$. A peak at 3,007 $cm^{-1}$ was assigned to the stretching of the C-H bond of aromatic rings; peaks at 2,914 and 2,851 $cm^{-1}$ were assigned to symmetric and asymmetric stretching of the C-H bond of alkyl groups, indicating a presence of alkyl chains; and a peak at 719 $cm^{-1}$ was assigned to bending of the C-H bond of aromatic rings. A higher transmittance signifies lower absorption, meaning that the number of bonds in this region was reduced. The major difference in spectra could also be seen in the range of 1,200–1,050 $cm^{-1}$, where two sharp peaks of the feedstock are not present in the spectra of bio-oils. These two peaks at 1,173 and 1,097 $cm^{-1}$ could be assigned to alcohols or phenols. Other peaks revealing the chemical composition of bio-oils were found at 1,709 $cm^{-1}$ (assigned to the stretching of C=O bond), which indicates the presence of carboxylic esters and acids. Additional peaks at 966, 1,119 and 1,240 $cm^{-1}$ were assigned to C–O bonds of acids and ethers (Yadav, 2013; Yin et al., 2010). The infrared spectra of our bio-oil were similar to the spectra reported by Yang et al. (2019) for biocrude, generated by HTL process of pork waste. However, a clear difference between our results and the spectra of the biocrude, derived from the pork waste, was in the presence of peaks that could be assigned to the presence of the N-H and C-N bonds in the latter material. Notably, our bio-oil has practically no nitrogen content.

## GC-MS analysis

A summary of the major components detected by GC-MS, in our bio-oil mixtures, with probabilities of their correct identification, is provided in Table 4. It is important to mention that not all compounds of the bio-oil are vaporized, detected by the MS detector and identified by the used library. Our bio-oils were found to contain methyl and ethyl esters of fatty acids, fatty acids, and even certain hydrocarbons. The presence of the fatty acids could be explained by the breakdown of triglycerides present in the AF. HTL of

triglycerides produces fatty acids in the oil phase and glycerol in the aqueous phase (*Biller & Ross, 2011*; *Déniel et al., 2016*; *Posmanik et al., 2017*; *Toor, Rosendahl & Rudolf, 2011*). This hydrolytic reaction occurs rapidly at temperatures above 280 °C (*Mills & McClain, 1949*). Although fatty acids are relatively stable up to 300 °C, *Déniel et al. (2016)* pointing out that some fatty acids can generate hydrocarbons, such as alkanes and alkenes *via* a decarboxylation reaction. The latter process can explain the presence of the two hydrocarbons in the list of the identified components (cholesta-7,14-diene and pentadecane, Table 3). *Posmanik et al. (2017)* also highlighted that high-temperature HTL process of lipid-rich biomasses, such as of meat waste, would contain primarily fatty acids and long-chain hydrocarbons.

Our list of compounds was similar to the results observed in the previous studies that explored the processing of different types of meat waste and lipid-rich biomass (*Leon et al., 2018*; *León, Marcilla & García, 2019*; *Yang et al., 2019*). The major difference between the meat waste and the biomass used in the present study was found in the lack of amides, N-heterocyclic, and any other nitrogen-containing compounds in our bio-oil. The decomposition of glycerol in near-critical water hydrolysis was reported to generate methanol, ethanol, and allyl alcohol, among other products (*Bühler et al., 2002*). When these alcohol groups interact with fatty acids of the triglycerides, esters of the fatty acids could be formed. It can explain the presence of fatty acid esters in our bio-oil. The abundance of fatty acids, their esters, and amides in the HTL bio-crude of lipid-rich biomass was previously reported (*Leonardis et al., 2013*). However, the mechanism of formation of methyl and ethyl esters was not mentioned and would require a detailed analysis.

## Comparison with marine fuels

Table 5 shows the comparison of fuel properties of the bio-oil produced in this work with two different marine fuels—Distillate Fatty Acid Methyl Ester (FAME) grade B (DFB) and Residual Marine A30 (RMB 30) based on *ISO (2017)*. The comparison made with marine fuels, as a promising application, was intended for use of our bio-oil directly or with minimal further processing. The specifications for marine engines have a broader scope of restrictions than petroleum products for road vehicles. The HHV of the bio-oil is 90–95% of the grade of marine fuels mentioned here. The viscosity and total acid number (TAN) are higher than the permissible limit for DFB but well within the range for RMB 30. *Tyrovola et al. (2017)* highlighted the requirement of low sulfur content in the future of marine fuels, which is planned to be 0.5%. As the sulfur content in the animal fat feedstock is very low, the %S in the bio-oil is merely 0.016%. This makes this bio-oil suitable for marine fuels now and in the future.

FAME in marine fuels was earlier restricted to 0.1% v/v as per EN ISO 8217:2012, but the standard issued in 2017 increased this limit to 7% in specific marine distillate grades (DF), thereby allowing the blending of FAME containing biofuels to these specific distillates (*Tyrovola et al., 2017*). Our bio-oil does have a certain quantity of FAME, as can be seen in the GC-MS results. However, it was not quantified. One of the main issues is the

**Table 4 Major compounds identified in the bio-oil obtained at reaction conditions of 330 °C, 60 min., and AF's solid load of 50%; using GC-MS, the US National Institute of Standards and Technology (NIST) library.**

| Elution time (min.) | Molecular formula | Component name | Identification probability (%) |
|---|---|---|---|
| 18.08 | $C_{15}H_{30}O_2$ | Tetradecanoic acid methyl ester | 92.58 |
| 25.33 | $C_{17}H_{34}O_2$ | Hexadecanoic acid methyl ester | 75.94 |
| 27.06 | $C_{18}H_{36}O_2$ | Hexadecanoic acid ethyl ester | 87.43 |
| 27.82 | $C_{10}H_{20}O_2$ | Decanoic acid | 75.35 |
| 28.49 | $C_{18}H_{34}O_2$ | Hexadec-9(e)-enoic acid ethyl ester | 72.6 |
| 34.63 | $C_{19}H_{38}O_2$ | Octadecanoic acid methyl ester | 48.95 |
| 36.01 | $C_{20}H_{38}O_2$ | 9-Octadecenoic acid ethyl ester | 74.12 |
| 36.97 | $C_{20}H_{38}O_2$ | Oleic acid ethyl ester | 51.85 |
| 44.55 | $C_{14}H_{28}O_2$ | Tetradecanoic acid | 88.32 |
| 46.00 | $C_{27}H_{44}$ | Cholesta-7,14-diene | 50.78 |
| 46.49 | $C_{15}H_{32}$ | Pentadecane | 84.89 |
| 47.28 | $C_{16}H_{32}O_2$ | 12-Methyltetradecanoic acid methyl ester | 70.87 |
| 48.67 | $C_{15}H_{30}O_2$ | Pentadecanoic acid | 88.73 |
| 51.22 | $C_{32}H_{62}O_3$ | Palmitic anhydride | 46.19 |
| 51.5 | $C_{18}H_{24}O$ | Estra-1,3,5(10)-trien-17β-ol | 51.56 |

**Table 5 Comparison of fuel properties of DFB[1], RMB 30[1] (ISO, 2017) & bio-oil from animal fat (AF), produced by HTL at 330 °C, 60 min, and 50% solid load.**

| Properties (Max. limit) | DFB[*,1] | RMB 30[1] | Bio-oil from AF |
|---|---|---|---|
| Viscosity ($mm^2$/s) | 11 (at 40 °C) | 30 (at 50 °C) | 21.69 (at 50 °C) |
| Density at 15 °C (kg/$m^3$) | 900.0 | 960.0 | 896.0–899.0 |
| Sulphur (% m/m) | 1.50 | [2] | 0.016 |
| Water (% V/V) | 0.30 | 0.50 | 6 |
| Total acid number (TAN) (mgKOH/g) | 0.5 | 2.5 | 1.46 |
| Fatty acid methyl ester (FAME) | 7.0 | [3] | [3] |
| Heating value (MJ·kg$^{-1}$) (Lin, 2013) | 42 | 40 | 37–38.84 |

Notes:
[*] The properties mentioned for DFB ("D" denotes distillate and "F" denotes FAME, B is a is distillate grade) and RMB 30 ( ) are maximum permissible limits for the fuels. Values for the bio-oil are measured values.
[1] Data of DFB and RMB ("R" denotes residual and "M" denotes marine fuel oil, B is a grade) 30 from (ISO, 2017).
[2] As requested by the supplier (ISO, 2017).
[3] Not quantified.

high moisture content in the bio-oil, which would require further processing of the bio-oil to make it suitable for marine uses.

## Economic analysis

In this section, we conduct an economic analysis of converting animal fat waste into bio-oil. The parameters in Table S3 relate to a lab-scale experimental system that converts 0.5 kg of animal fat waste into 0.133 kg bio-oil as the main product, while the energy needs of the process can be supplied by combusting a portion of the solid residue remaining after the animal fat processing (thus incurring zero energy variable cost of the process). As a

result, Table S3 shows no external heat supply and demonstrates a sustainable HTL process. As the water is an input and output (assuming full separation of the aqueous phase products) in our production process, its net balance of 0.06 kg can be considered pure water and valued as such. The energy consumption in extracting the bio-crude oil and other products from the complete HTL output has not been considered here and is needed in the future to fulfill the study economics.

We assumed that the price of animal fat waste equals the price of animal fat; the latter is closely correlated with the prices of vegetable oils (Malins, 2017). We set the price of animal fat waste in the baseline to 0.32 euros per kilogram, which corresponds to the price of category 1 and 2 animal fat (USDA, 2021). In sensitivity analysis, we set the upper bound of the price of animal fat waste to 0.36 euros per kilogram, which corresponds to category three fat. We set the lower bound of the (Malins, 2017) price to zero, recognizing the possibility that some processors might be willing to give the fats away to dispose of them in case of missing infrastructure for further processing in their vicinity.

In this economic analysis, the produced bio-oil is assumed to be used to produce biodiesel, and its price is assumed to be the same as the price of used cooking oil, namely, 1.00 euro per kilogram in the baseline case (this reflects the Rotterdam FOB price of used cooking oil between 2017 and 2020 (Eurostat, 2021)). Used cooking oil was chosen as a reference because, akin to animal fat waste, it is a feedstock for second-generation biodiesel (unlike, for example, rapeseed oil). Finally, the baseline price of water is set to be 1.5 euros per cubic meter (0.0015 euros per liter), which is at the lower edge of prices in Germany, England, France, The Netherlands, Austria, and Poland (Civity Management Consultants, 2018).

Part of the leftover solid fat residue (Table S3) is assumed to be valorized through combustion for generating electricity. Given that the LHV of 1 kg of solid fat residue is 36 MJ and the conversion efficiency to electricity is 30%, the total amount of electricity generated from the 0.337 kg of solid fat residue (the last row in Table S3) is 1.01 kWh. For the scaled-up process below, we assumed the possibility of 50%, 60%, 70%, 80%, and 90% heat utilization from combusting the solid residue. The baseline price of electricity is considered to be 0.10 euros per kWh and corresponds to the mean electricity prices for non-household customers (including non-recoverable taxes) as provided by Eurostat (2021).

The input and output values in Table S3 coupled with the corresponding prices, result in the total variable cost of 0.16 euros against revenue of 0.23 euros (with electricity co-generation). These calculations do not consider the annualized capital cost per liter of bio-oil and per kWh of electricity or other operating costs (e.g., labor).

We also investigated the economic effects of scaling up the production process in Table S3 to 50,000 kg of animal fat per day. All variable inputs and outputs from Table S3 were prorated. The ensuing profit margin (i.e., revenues minus variable costs) equals 7,411 euros per day in case of bio-oil only, increasing to 7,865 euros for 50% utilization of the solid residue (60%: 7,956 euros, 70%: 8,047 euros, 80%: 8,137 euros and 90%: 8,228 euros). If animal fat waste is obtained free of charge, the above profit margins increase to 23,411 euros per day (0% recuperation) and 23,865 euros per day (50% heat recuperation).
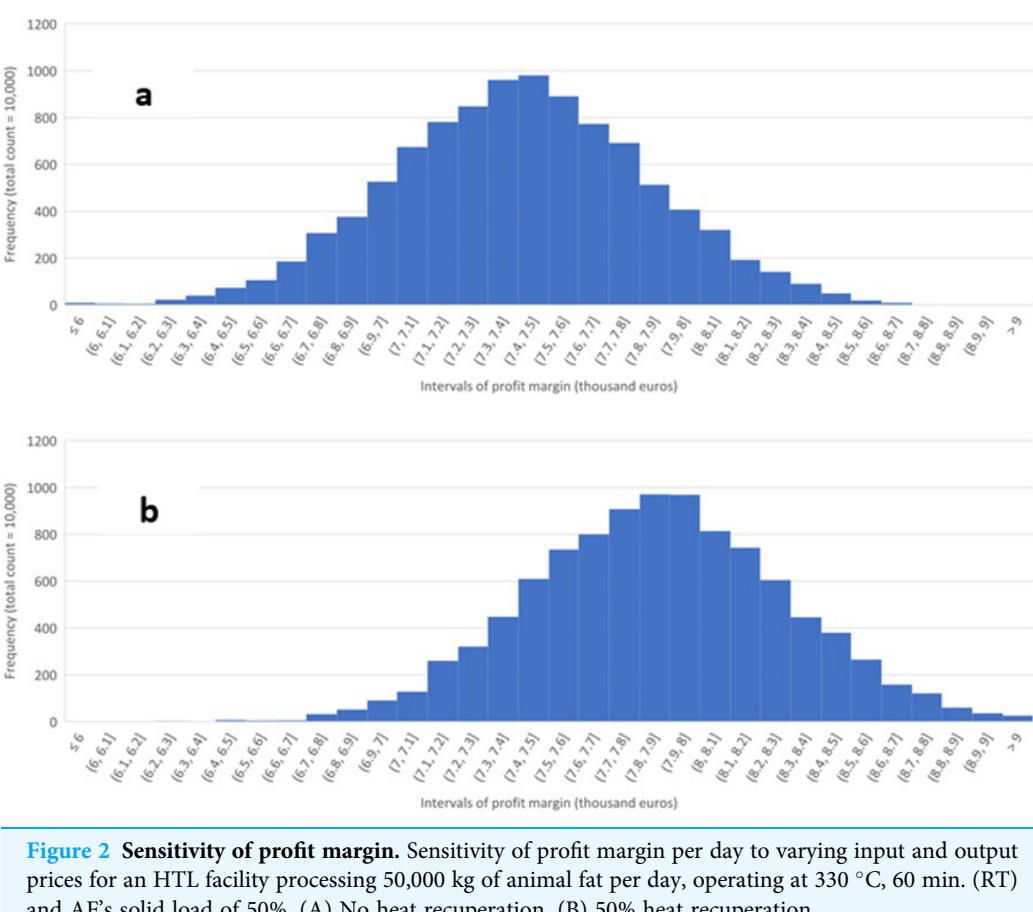

**Figure 2 Sensitivity of profit margin.** Sensitivity of profit margin per day to varying input and output prices for an HTL facility processing 50,000 kg of animal fat per day, operating at 330 °C, 60 min. (RT) and AF's solid load of 50%. (A) No heat recuperation, (B) 50% heat recuperation.

Considering the above conservative estimate of the profit margin of 7,411 euros per day and assuming the facility works daily, the profit margin per year would be 2.71 million euros. Suppose the interest rate is four percent. Then the sum of the discounted stream of benefits over 20 years would be 36.76 million euros. An investor would need to use this value as a threshold for comparison of any initial capital cost (and the sum of discounted operation cost, *e.g.*, labor, maintenance) to determine if the investment is likely to break even over 20 years.

Figure 2 summarizes the results of the sensitivity analysis concerning varying input and output prices, holding the technological parameters at their baseline levels. Each histogram is based on the outcomes of 10,000 Monte Carlo simulations in which the prices were randomly and independently drawn from a PERT distribution whose parameters were determined by the minimum, baseline, and maximum values of the prices above. The minimum and maximum prices of electricity, bio-oil, and water in the sensitivity analysis are assumed to be 30 percent below and above the baseline prices.

Figure 2A depicts the results for no-heat recuperation, and Fig. 2B assumes 50% heat recuperation. The horizontal axes display the intervals (bins) of profit margin in thousands of euros per day, while the vertical axes measure how many simulated profit margins (out

of 10,000) correspond to a given interval. The figures demonstrate the economic importance of recouping at least some share of the wasted heat.

The average share of bio-oil and electricity in the revenues is 56.8% and 43.28% for no heat recuperation; 55.7% and 44.2% for 50% heat recuperation, and 54.9% and 45.1%, for 90% recuperation. These results demonstrate the growing economical share of the electricity in the revenues as commercial production becomes more energy-efficient. The contribution of water is negligible. It is recommended that in future research, the cost of removal of water and upgrading of bio-crude be taken into account.

# CONCLUSIONS

This study reveals that the rendered bovine fat could be converted into bio-oil by a hydrothermal liquefaction process. The formed bio-oil has an HHV of $38.5 \pm 0.16$ MJ·kg$^{-1}$, which is 90–95% of marine-grade fuels. The process does not require additional energy, reduces waste handling costs and fulfills the demand for carbon neutral marine fuel. Furthermore, the nitrogen and sulfur contents of our bio-oil were found to be negligible, suggesting its prospective use for future production of marine biofuel conforming with the current strict limiting regulations of $NO_x$ and $SO_x$ emissions. The economics of the process of conversion of animal fat to bio-oil, and the generation of electricity from combusting the solid residue were also analyzed. The revenue analysis indicates that the production process could produce a positive profit margin (excluding the capital cost), even without utilizing the solid residue. The heat recuperated from combusting the solid residue further improves the economics of the production process. With a higher rate of heat recuperation, electricity generation would progressively gain a greater share in revenues, relative to the bio-oil.

## Funding
This work was supported by the Israel Ministry of Energy (#219-11-138). Dušan Drabik received financial support from the Slovak Research and Development Agency under contract No. APVV-19-0544 and from the Operational program Integrated Infrastructure within the project: Demand-driven research for the sustainable and innovative food, Drive4SIFood 313011V336, co-financed by the European Regional Development Fund. The funders had no role in study design, data collection and analysis, decision to publish, or preparation of the manuscript.

## Grant Disclosures
The following grant information was disclosed by the authors:
Israel Ministry of Energy: #219-11-138.
Slovak Research and Development Agency: APVV-19-0544.
Operational Program Integrated Infrastructure within the Project: Demand-driven Research for the Sustainable and Innovative Food: Drive4SIFood 313011V336.
European Regional Development Fund.

## Competing Interests

The authors declare that they have no competing interests.

## Author Contributions

- Efraim Steinbruch conceived and designed the experiments, performed the experiments, analyzed the data, prepared figures and/or tables, authored or reviewed drafts of the article, and approved the final draft.
- Siddaq Singh conceived and designed the experiments, performed the experiments, analyzed the data, prepared figures and/or tables, authored or reviewed drafts of the article, and approved the final draft.
- Maya Mosseri performed the experiments, analyzed the data, prepared figures and/or tables, authored or reviewed drafts of the article, and approved the final draft.
- Michael Epstein performed the experiments, analyzed the data, prepared figures and/or tables, authored or reviewed drafts of the article, and approved the final draft.
- Abraham Kribus conceived and designed the experiments, analyzed the data, authored or reviewed drafts of the article, and approved the final draft.
- Michael Gozin conceived and designed the experiments, analyzed the data, prepared figures and/or tables, authored or reviewed drafts of the article, and approved the final draft.
- Dušan Drabik performed the experiments, analyzed the data, prepared figures and/or tables, authored or reviewed drafts of the article, and approved the final draft.
- Alexander Golberg conceived and designed the experiments, analyzed the data, prepared figures and/or tables, authored or reviewed drafts of the article, and approved the final draft.

## Data Availability

The raw data is available in the Supplemental Files.

## Supplemental Information

Supplemental information for this article can be found online at http://dx.doi.org/10.7717/peerj.16504#supplemental-information.

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
