# Peer review of "Waste animal fat with hydrothermal liquefaction as a potential route to marine biofuels"

_PeerJ, doi:10.7717/peerj.16504_

## Round 0.1 · original submission · Major Revisions

Thank you for your submission to PeerJ. The work that you have presented is very timely and relevant to novel approaches for production of biofuel. The two reviews were quite critical of the present manuscript and although not in complete agreement with all of the comments, I do feel the manuscript would benefit from a major revision focused on a thorough discussion and hypotheses concerning the results of the various feedstock loads, the three phases found, and their chemical constituents. Providing explanations for observed processing outcomes would give a more clear view of the results and their significance within the biofuel research area. For example, the low load systems have diminished bio-oil production. Is this due to increase in gasification? It would be useful to include gas analysis data in the manuscript with a bar chart of percentages in the main manuscript. All of the results presented should have an expanded discussion. Showing that a high feedstock load, without catalyst but under fairly extreme HTL conditions, is an intriguing result. This may direct researchers using HTL to process other materials at higher temperatures and higher feedstock loads. It would be useful to examine the literature in these areas to reflect the similarities and differences in outcomes, especially in the non-catalytic processing systems and in the processing of triglyceride systems, in expanding the Discussion and Analysis. Another specific issue that should be addressed is how all of the bio-oil can be removed without a solvent which isn’t clear in the manuscript. What percent of this is remaining in the vessel and how is this assessed? Lastly, there are a number of grammatical errors throughout the manuscript that should be corrected.

The reviewers both made several suggestions given below that should be specifically addressed and discussed in the revised manuscript.

Reviewer 1. As a result, the experiments were conducted at unusual severity conditions at extremely high solid contents as high as 75%. With 300+ C temperature and retention time as long as 120 minutes, the reaction condition s were so severe, resulting in that fat (lipid primarily) was depolymerized into something else (likely acid) rather than biocrude oil. Typical HTL reactions are ~20% solid content, and reaction HTL severity factor no more than 7.5. A good literature review could help to improve the study --- There are a wealth of literature on HTL for various biomass that could be insightful for this study.
The experimental design should be clearly explained, and the biochemical properties of the animal fat should be analyzed and presented.

Reviewer 2. The authors have provided more of the reasons/observations from the previously reported work. An explanation about the current work is much less.
Therefore, this manuscript needs major revisions along with in-depth analysis before publication. The specific comments are:
• For TGA analysis, 2 mg samples were used. Is this amount good enough to get concrete results. Have the authors tried 10-20 mg samples for analysis?
• Page 39: Table 1: please correct “temperature impact” to impact of retention time”.
• Table 4: please spell out shor

t forms using DFB, RMB etc.
• How about the solid residue left, its analysis, uses, etc.?
• English language needs improvement.

Reviewer 1 ·

Basic reporting

This manuscript reports a study of converting waste animal fat with hydrothermal liquefaction as a potential route to marine biofuel. While the topic is important, the experiment was poorly designed and the results could be misleading. The manuscript is not acceptable for the following reasons:
The authors appear not familiar with the hydrothermal liquefaction literature and technology. As a result, the experiments were conducted at unusual severity conditions at extremely high solid contents as high as 75%. With 300+ C temperature and retention time as long as 120 minutes, the reaction condition s were so severe, resulting in that fat (lipid primarily) was depolymerized into something else (likely acid) rather than biocrude oil. Typical HTL reactions are ~20% solid content, and reaction HTL severity factor no more than 7.5. A good literature review could help to improve the study --- There are a wealth of literature on HTL for various biomass that could be insightful for this study.
The experimental design should be clearly explained, and the biochemical properties of the animal fat should be analyzed and presented.

Experimental design

Experimental design is poor.

Validity of the findings

Results could be misleading

Reviewer 2 ·

Basic reporting

• English language needs improvement.

Experimental design

The experimental design is described with sufficient details and information.
Research question defined is not up to mark.

Validity of the findings

The authors have provided more reasons/observations from the previously reported work. An explanation about the current work is much less. The work reported does not seem to be novel.

Additional comments

Major revisions required

Annotated reviews are not available for download in order to protect the identity of reviewers who chose to remain anonymous.

---

## Round 0.2 · Minor Revisions

Thank you for submitting your revised manuscript. One of the original reviewers did not respond to requests to rereview and Another was added. Please look at the two reviewers comments to guide minor revisions of the manuscript. Please also pay close attention to some typos and missing information by carefully going through the manuscript after addressing the reviewer comments. One of example of this is reference in the manuscript (Line 225 H & McClain, 1949a)

Reviewer 2 ·

Basic reporting

See below

Experimental design

See below

Validity of the findings

See below

Additional comments

The manuscript is improved in revised version and the comments raised were addressed. Therefore, this manuscript can be accepted for publication after correcting following points:

• Lines 249-251: Please site reference used for oxygen content determination?
• Lines 522: Correct “Combuting” to “combusting”.
• English language needs improvement.

·

Basic reporting

There seems to be too many references that are not applicable to the study of bovine fat HTL.

Moisture content really needs to be added
- Initial feed stock
- Post HTL
- Post 30min initial centrifuge
- Post 30min final centrifuge

Nice amount of background information, but it would seem that the focus really needed to be on how bovine fat differs. Miscibility being one focus area.

Experimental design

Miscibility of the feedstock and the decision whether to cross the 320 deg Celsius gap was a key research question. More insight from the researchers would have valuable for future researchers.

Moisture content, throughout the process, would be results desired, but not provided.

Validity of the findings

250 ml reactor size challenges the validity of the results. The limited gas portion of the HTL output points to challenges with experimental setup.

Once again claims of mechanical centrifuge without requiring solvents and economic viability without addressing drying the solid residual challenge the validity of the economic analysis.

I believe that economic analysis of the entire process is critical for the success of HTL and biomass industry. I commend the authors for attempting. With the moisture content question answered, this economic insight would be very beneficial to those engaged in the field of bovine waste management.

---

## Round 0.3 · accepted · Accept

Thank you for resubmitting your manuscript and addressing the reviewer comments completely. I have attached a PDF with some editorial corrections that will be made in the final proof with your acceptance.

·

Basic reporting

No further comments.

Experimental design

No further comments.

Validity of the findings

No further comments.

Additional comments

No further comments.